# Cold-Chain-Food-Related COVID-19 Surveillance in Guangzhou between July 2020 and December 2022

**DOI:** 10.3390/foods12142701

**Published:** 2023-07-14

**Authors:** Zongqiu Chen, Xiaoning Li, Jinhua Zhou, Tengfei Zhou, Tianji Lin, Conghui Xu, Jianhai Yu, Kuibiao Li, Zhoubin Zhang, Wei Zhao

**Affiliations:** 1BSL-3 Laboratory (Guangdong), Guangdong Provincial Key Laboratory of Tropical Disease Research, School of Public Health, Southern Medical University, Guangzhou 510515, China; harpul@sina.com (Z.C.); chienhai@163.com (J.Y.); 2Guangzhou Center for Disease Control and Prevention, Guangzhou 510440, China; ningqzone@163.com (X.L.); 18898534067@163.com (J.Z.); flyingtank@126.com (T.Z.); lintianji@126.com (T.L.); yeast-27@163.com (C.X.); kuibiao@outlook.com (K.L.)

**Keywords:** COVID-19, SARS-CoV-2, cold-chain food, centralized supervision warehouse system, outbreak

## Abstract

Objective: To monitor severe acute respiratory syndrome coronavirus 2 (SARS-CoV-2) RNA contamination in samples linked to imported cold-chain food and assess the situation from the implementation of a centralized supervision warehouse system in Guangzhou, Guangdong Province, China. Methods: Swabs of workers and frozen-food-related samples were collected between July 2020 and December 2023 in Guangzhou, Guangdong Province. SARS-CoV-2 RNA was extracted and analyzed by a real-time quantitative polymerase chain reaction using the commercially available SARS-CoV-2 nucleic acid test kit. The risk level and food source were monitored simultaneously. Results: A total of 283 positive cold-chain events were found in Guangzhou since the first cold-chain-related event of the coronavirus disease 2019 pandemic was identified in July 2020. Most positive samples were a low-to-medium risk, and the cycle threshold value was >30. No live virus was detected, and no staff came into direct contact with a live virus. In total, 87.63% of positive events were identified through sampling and testing at the centralized food warehouse. Conclusion: Cold-chain food has a relatively low risk of transmitting SARS-CoV-2. Centralized food storage can be used as an effective method to control this risk, and this measure can also be used for other food-related, contact-transmitted diseases.

## 1. Introduction

Some laboratory studies have shown that severe acute respiratory syndrome coronavirus 2 (SARS-CoV-2) remains highly stable under refrigerated (i.e., 4 °C) and freezing (i.e., −10 to −80 °C) conditions in fish, meat, poultry, and swine skin for 14–21 days; therefore, contaminated cold-stored food may present a systematic risk for the transmission of SARS-CoV-2 between countries and regions [1,2,3]. Research evidence suggests that cold-chain transportation in the frozen food industry can cause the recurrence of coronavirus disease 19 (COVID-19) cases at destination [4,5,6,7,8]. Although the likelihood of food-to-human transmission is considered lower than for other means of transmission such as through respiratory droplets, cold-chain transportation should not be neglected as a risk factor, given the large volumes of refrigerated food that are transported across different countries and regions [9]. The COVID-19 outbreaks related to cold-chain food in Xinfadi’s agricultural wholesale produce markets of Beijing [10,11], Qingdao of Shandong Province [12], and Dalian of Liaoning Province [4] have prompted the Chinese government to strengthen the prevention and control of COVID-19 transmission through cold-chain food.

Before the COVID-19 outbreak, China had implemented a food safety supervision and management system and formulated the National Food Safety Law in 2009 to manage food safety at the highest level in the form of national legislation. Under the framework of the National Food Safety Law, a series of national food safety standards and more detailed regulations include very specific management requirements for food production, processing, circulation, and consumption. Guangzhou port is one of the ports that receives the largest amount of imported food in China. In Guangzhou, the customs department is responsible for the supervision and management of the entry of imported cold-chain food. Before food is imported, food manufacturers must register with Guangzhou customs and provide the required certification materials. For example, imported meat products must be accompanied by official certificates, and imported pre-packaged food should carry a Chinese label that meets the requirements. Upon importation, customs officials conduct a sampling inspection and risk monitoring of imported food that includes the monitoring of 180 items of chemical contaminants and microorganisms.

Customs issues an inspection and quarantine certificate for the imported goods after the conformity assessment, and the imported food can then be sold and used. After import, Guangzhou Province requires that manufacturers keep a record of food import and sales; food name, specifications, quantity, production date, production batch number or import batch number, and shelf life; names of overseas exporters and buyers; addresses and contact information; delivery date; and other information. Manufacturers are also required to save relevant vouchers.

To prevent the transmission of COVID-19, Guangzhou customs began implementing SARS-CoV-2 nucleic acid testing for imported cold-chain food in March 2020 and comprehensive nucleic acid testing after June 2020. Since July 2020, Guangzhou Municipality has established a prevention and control system for imported cold-chain food. A centralized supervision warehouse system was established in January 2021. The following associated policies were subsequently implemented: (1) each batch of imported cold-chain goods must be individually sampled, with samples taken of the inner and outer packaging of each product; and (2) the centralized closed-loop management of employees and positive goods should be managed promptly.

In this study, we analyzed the data from the monitoring program for imported cold-chain food that aimed to screen for SARS-CoV-2 RNA contamination in samples linked to imported cold-chain food and assessed the situation since the implementation of the centralized supervision warehouse system in Guangzhou, Guangdong Province, China. The study results will guide the improvement of cold-chain food management and decrease food-related infectious diseases that are transmitted through food contamination.

## 2. Materials and Methods

### 2.1. Sampling Methods

The establishment of a centralized supervision warehouse system for cold-chain food in Guangzhou required that each batch of imported cold-chain food be sent to a centralized supervision warehouse for disinfection and sampling before the batch entered market circulation. The sampling quantity was in proportion to the total number of the same type and batch of goods.

The sampling method was in accordance with the Notice on Issuing Technical Guidelines on the Classification and Disposal of Cold-Chain Food for COVID-19 Prevention and Control, which was published by The Joint Prevention and Control Mechanism of The State Council for COVID-19 [13]. The sampling principle should follow the principle of randomness and representativeness during the normal epidemic prevention and control period and should focus on cold-chain food and associated packaging from high-risk areas. The number of samples followed the formula 100 × 3% + (500 − 100) × 2% + (1000 − 500) × 1% + (N − 1000) × 0.5%, in which N was the quantity of cold-chain food. A non-inactivated virus sampling tube was used for sampling, and adequate personal protection and aseptic procedures were used to prevent cross-contamination during sampling. The samples must have been transported to the laboratory at low temperatures within 4 h for testing, and whole-genome sequencing and virus isolation culture were performed for high-viral-load samples.

Accordingly, the outer packaging of each sample of imported cold-chain food was checked before storage, the inner packaging and product surface of imported cold-chain food in each batch was inspected in accordance with the method of proportional sampling, and the surface of samples were confirmed to be in a non-frozen state before sampling and disinfection. Each sampling tube contained five or 10 swabs, and each swab was applied to the surfaces of 5–10 boxes of goods. Each box of goods was tested on at least two sides. In addition, environmental sampling covered the chain from cold storage, transportation, and processing to the sale of imported cold-chain food. Environmental sampling also encompassed cold-chain logistics enterprises, farmers’ markets, food production enterprises, shopping malls, supermarkets, and other locations. The surface of a table that had the most frequent contact with personnel or goods was selected for sampling. Samples were transported to the testing agency at a temperature of 4 °C within 12 h of sampling. Sample transportation and testing complied with relevant regulations on biosafety. If imported cold-chain food, the environment, or employees were found to be positive for COVID-19 nucleic acid, the food was discarded, in accordance with relevant regulations.

### 2.2. Disinfection Process

Each batch of newly arrived cold-chain food was disinfected before being stored in the warehouse. Peracetic acid (0.1~0.2%), hydrogen peroxide (3%), 1000 mg/L chlorine-containing disinfectant, or 2000 mg/L quaternary ammonium salt disinfectant were used to fully wipe, spray, or soak goods for 30 min. All of the six sides of the items were disinfected, and surface ice was removed before disinfection. Disinfection supplies, such as disinfectants and disinfection instruments, were selected in accordance with national health standards. Selected disinfection products were registered on the national online information platform.

After handling a batch of goods in the working area with the appropriate tools, 500–1000 mg/L of chlorine-containing disinfectant, peracetic acid (0.1–0.2%), or hydrogen peroxide (3%) was applied to the surface inside the transportation container, the floor of the working area, and equipment such as carts, shelves, forklifts, and related tools, utensils, and containers. The treatment time was 30 min; the disinfected surface was then washed with clean water to remove residual disinfectant. Cold-storage facilities, refrigerators, and freezers were sterilized weekly. Any visible contaminants on tables and relevant tools and appliances and in the work area environment or cold-storage refrigerators were cleaned before disinfection.

When cold-chain goods entered market circulation, comprehensive preventive disinfection was performed in accordance with the principle of “whoever unpacks must disinfect” and on the premise of “not affecting food safety and quality.” Outer packaging was disinfected to ensure food safety.

Disinfectants that were used in antifreeze conditions were verified using simulated field tests. The disinfection unit established ledgers to ensure the detailed recording—including disinfection date, personnel, location, disinfection object, disinfectant name, concentration, action time, and the amount of disinfectant used—of the disinfection of goods and the environment. Relevant data and records were retained for a specified period.

### 2.3. Management of Staff, Daily Protection, and Work Requirements

The staff in the centralized supervision warehouse were managed by closed-loop management and underwent daily nucleic acid testing and health monitoring. If test results from the daily monitoring of staff were abnormal, the staff stopped working immediately and were isolated. Secondary self-protection was adopted in the work that contact with cold-chain-food directly, and supervisors who were responsible for staff protection provided timely reminders to staff. The staff performed relevant work in a specified area in accordance with a fixed process. Staff in the centralized supervision warehouse were responsible for six activities (relative concentration, on-site supervision, electronic tracing, piece inspection, piece disinfection, and qualified delivery) and three special measures (special channel purchase, area storage, and area sales) related to imported cold-chain food. The storage and sale of cold-chain food with other food was prohibited.

### 2.4. Procedures for Nucleic Acid Testing, Positive Reporting, and Handling Outbreaks

The testing agency tested samples from the centralized supervision warehouse, and the test results were reported to the Guangzhou Municipal Administration of Market Supervision and the Guangzhou Center for Disease Control and Prevention (GZCDC). The GZCDC then issued a weekly assessment report to the Guangzhou Health Commission and Guangzhou Municipal Administration of Market Supervision and proposed targeted recommendations. All positive specimens were sent to the Guangzhou GZCDC for reexamination within 2 h, and the reexamined samples were then sent to the Guangdong CDC within 24 h for virus isolation and gene sequencing. Concurrently, several associated departments were prepared to respond to emergencies by collecting information, screening and controlling personnel, conducting a patriotic health campaign, identifying prevention and control units, expanding sampling, tracing the flow of contaminated products, conducting terminal disinfection, disposing of nucleic acid positive products, and releasing information (Figure 1 and Figure 2). If required, an incident risk assessment was then conducted by the GZCDC and the result was reported to relevant departments, thus concluding the emergency response.

To meet the needs for rapid disposal of cold chain events the National Health Commission issued new technical guidelines for the classification and disposal of cold-chain food on 2 June 2022. The basis for risk assessment and the requirements of hierarchical disposal were moderately changed in the new guidelines. The main changes were the classification of risk levels; no changes were made to the treatment of each type of risk.

### 2.5. Laboratory Methods

Laboratory detection methods included nucleic acid detection using polymerase chain reaction (PCR), virus whole-genome sequencing, and isolation. The extracted sample RNAs were divided into separate packages and analyzed using real-time quantitative PCR with a commercially available SARS-CoV-2 nucleic acid test kit. The TaqMan probe-based kit (Wuhan Mingde Biotechnology Co., Ltd. (Wuhan, China), Guangzhou Da’an Gene Co., Ltd. (Guangzhou, China), and Jiangsu Shuoshi Biotechnology Co., Ltd. (Shanghai, China)) was designed to detect the ORF1ab and N genes of SARS-CoV-2 in one reaction using the procedure described by the Joint Prevention and Control Mechanism of the State Council of the People’s Republic of China. All test kits for SARS-CoV-2 nucleic acid extraction sequencing, isolation, and real-time quantitative PCR analysis were approved by the State Food and Drug Administration of China.

### 2.6. Data Collection and Analysis

The data were periodically collected from a centralized supervision warehouse and were analyzed by the Guangzhou Center for Disease Control and Prevention. The main analysis recorded sample size, risk classification, and the COVID-19 positive rate following the establishment of the centralized supervision warehouse. The risk classification of positive samples, risk grading, cycle threshold (Ct) value, and other aspects were analyzed. Virus isolation and whole sequencing were conducted for high-risk samples. The results of outbreak management were also analyzed.

## 3. Results

### 3.1. Temporal Trend and Sources of Positive Cold-Chain Events

A total of 283 positive cold-chain events were found in Guangzhou since the first cold-chain-related event of the COVID-19 pandemic was identified in July 2020. The number of positive cold-chain events increased annually, with 220 (77.74%) of the events observed in 2022. Positive cold-chain events occurred most frequently in autumn and winter, whereas fewer events occurred from March to April. The highest number of positive events (40; 14.13%) occurred in October 2022. Positive samples were mainly associated with meat products (170; 60.07%). The numbers of events that involved aquatic products, fruit products, and other food products were 65 (22.97%), 17 (6.01%), and 26 (9.19%), respectively. The positive results in the cold chain across food were consistent with the distribution of overall positive events (Figure 3).

The products associated with positive events were from Asia (91; 32.16%), South America (71; 25.09%), North America (71; 25.09%), Europe (60; 21.2%), and Australia (52; 18.37%). Australian imports had the highest number of positive samples (4.75) per positive event, followed by imports from Asian countries (3.6 per event; Table 1).

### 3.2. Location of Positive Events along the Cold Chain

Positive events were found during interception at centralized supervision cold storage or downstream cold storage and during the interception of freight or smuggled products. The most positive events were observed during interception at centralized supervision cold storage (87.63% of positive events and 95.91% of positive samples), which reduced downstream communication risks in the sales process of cold-chain food. All of the imported cold-chain food was classified into four categories (fruits and fruit products, aquatic products, meat products, and other food). The most positive events were observed in meat products (61.84% of positive events and 56.53% of positive samples). The number of positive samples (3.50) per positive event was highest in the centralized supervision warehouse (Table 2).

### 3.3. Location of Positive Cold-Chain Events

A total of 272 positive events and 886 positive samples were observed. Positive events and samples were found in product packaging, internal packaging, product surfaces, and environmental surfaces. In some events, two types of positive results were found. Four (1.4%) events were positive for outer packaging and inner packaging, one (0.35%) event was positive for outer packaging and the environment, and three (1.06%) events were positive for inner packaging and environmental surfaces. The positive events and samples were mainly associated with product packaging. Meat products accounted for 62.13% and 63.64% of the positive results associated with outer and inner packaging, respectively (Table 3).

### 3.4. Risk Classification of Positive Cold-Chain Events

According to the cold-chain event risk classification strategy, 226 events were considered low-risk events (79.86%), whereas medium- and high-risk events accounted for 19.79% and 0.35% of events, respectively. The first medium-risk event was reported in January 2021; medium-risk events were reported monthly after August 2021. Only one high-risk event was reported in the regulatory cycle; the product associated with the event was chicken feet imported from Brazil in November 2021. The Ct value of the positive specimen was between 27.23 and 38.01. A total of five positive specimens were found in 225 samples that were collected using preliminary testing and expanded sampling (Table 4).

### 3.5. Centralized Supervision of Warehouse Environment and Nucleic Acid Testing of Personnel

Five centralized supervision cold-storage facilities in Guangzhou were distributed across four districts (two facilities in Huangpu District, and one each in Nansha, Panyu, and Baiyun Panyu). A total of 13,094 high-risk individuals and 75,962 low-risk individuals were monitored, and none were found to be positive. A total of 972,738 cold-chain food samples were monitored, and 904 were found to be positive (positive rate = 0.09%). A total of 14,896 environmental samples were monitored, and six were found to be positive (positive rate = 0.04%; Table 5).

## 4. Discussion

Given that the SARS-CoV-2 virus can survive in low temperatures for a long period, imported cold-chain food poses a risk of infection across all cold-chain links, including processing, transportation, storage, downstream market flow, and consumption. The risk of transmission is highest in trans-regional transportation and downstream market consumption [5,8]. During cross-regional transportation, drivers who transport cold-chain food are exposed to cold-chain food packaging. Inadequate protection during transportation and the rubbing of the face, eyes, mouth, nose, and other exposed body parts with bare or gloved hands places drivers at substantial risk of COVID-19 infection [14,15]. During handling, porters touch frozen or refrigerated meat with bare hands, and product packaging workers share work clothes and live and dine with others after leaving their posts, causing the spread of the virus within a small area [12,16]. Moreover, the continuous low temperature in the downstream cold chain and the market increases the risk of COVID-19 transmission from cold-chain practitioners to ordinary residents [3,5,12,17].

Disinfection plays an important role in sealing off the cold chain as a transmission route for SARS-CoV-2 but faces many challenges. For example, disinfection of the outer packaging of cold-chain food with disinfectants is influenced by low temperatures—especially those of the ice that forms on the outer packaging. Therefore, achieving optimal disinfection is difficult. Furthermore, the erosion of food and external packaging affects the quality of food and packaging [18,19]. As cold-chain food can transmit many types of contact-borne diseases, low-temperature disinfection technology should remain one of the main research directions of cold-chain food risk control [11].

Since July 2020, the city of Guangzhou has comprehensively operationalized the centralized supervision warehouse system for imported frozen food and has implemented centralized testing, disinfection, and supervision of imported frozen food that enters the city and each element of relevant warehousing. Guangzhou gradually introduced electronic traceability, implemented classified management, and centralized the residences, with high-frequency monitoring and whole-process supervision of the personnel who worked in the centralized supervision warehouse. The city’s production and import of cold-chain food units were also regularly supervised [10,12]. We will continue to comprehensively monitor environmental products and workers. The results of this study showed that the overall positive detection rate was low in cold-chain food and the associated environment and personnel. The majority of positive samples were of low-to-medium risk, and the Ct value was greater than 30. No live virus was detected during the supervision period, no positive person on the staff was identified as being in direct contact with SARS-CoV-2, and no positive person was identified during the closed-loop management period. During centralized management, strong personal protection, environmental and food disinfection, and the concentrated residence of personnel may play important roles in reducing the risk of COVID-19 transmission through cold-chain food [10]. In May 2021, the number of centralized supervision warehouses in Guangzhou was reduced from 12 to 5 when a more centralized supervision system was implemented. With the adjustment of China’s epidemic prevention and control policy in December 2022, the centralized supervision warehouses ceased operation; however, their use remains authorized to support responses to future large-scale epidemics. Therefore, the prevention and control measures of the cold chain can be further optimized. Large-scale sampling is no longer required, but continuous sampling and the monitoring of cold-chain products should be maintained, and the range of virus spectrum monitoring can be extended to other pathogens such as cholera, norovirus, and other pathogens that cause infectious diarrhea [9,14,20].

The results of this study showed that the considerable increase in positive cold-chain time in 2022 compared with those in 2020 and 2021 was related to the growing global COVID-19 epidemic and improved supervision [15,21]. No significant difference was observed across regions in the positive detection rate of products, suggesting that the number of positive events was correlated with the volume of imported goods [11,21,22,23]. In the case of the global COVID-19 pandemic, the risk of importing contaminated goods was the same across all regions. The proportion of positive events identified through an interception in the centralized supervision warehouse was 87.63%, suggesting that the centralized supervision warehouse system played an important role in preventing risk spillover. A relationship may exist between the occurrence of cold-chain events that involve meat products and the total number of cold-chain goods. Some studies have shown that cold-chain products are not themselves associated with COVID-19 transmission risk and that the risk primarily derives from the processing and transportation of cold-chain products [24,25,26].

Focusing attention on the health risks caused by cold-chain food remains necessary even in the absence of COVID-19. To reduce the risk of infection caused by contact with all pathogens including COVID-19, we suggest that additional focus be applied to the health risks of cold-chain food after the COVID-19 pandemic [7]. Future research and practice should focus on several strategies. First, cold-chain food safety supervision during product processing, storage, and transportation must be strengthened. During these stages, direct contact with food should be minimized as much as possible, and antibacterial and bacteriostatic approaches or disinfection should be used on packaging surfaces. Second, publicity and education must be reinforced to remind consumers to strengthen personal protection and disinfection when buying and preparing cold-chain food to avoid infection caused by unpacking and cooking. Finally, focus must be directed to research on disinfection measures for low-temperature cold-chain food. At present, only a few disinfection methods are used for low-temperature cold-chain food; however, these methods are imperfect and mainly result in poor disinfection and damage to product and packaging performance.

Our study has some limitations. First, samples that taken in the market or during traffic interception were detected by a testing company, of which we were informed only when positive samples were detected; thus, the whole sample size was not obtained, resulting in the actual positive rate being lower than in our monitoring data. Second, during the COVID-19 epidemic, the management of cold-chain food and personnel by the centralized supervision warehouse was based on several prevention and control measures, and its effect on preventing the infection risk from cold-chain imports was difficult to quantitatively evaluate. Finally, throughout the monitoring period, the SARS-CoV-2 virus had not been successfully sequenced and isolated from positive samples; therefore, direct evidence of transmission from contaminated cold-chain food to humans was lacking.

## 5. Conclusions

Cold-chain food has a relatively low risk of transmitting SARS-CoV-2. Centralized food storage can be used as an effective method to control this risk, and this measure can also be used for other food-related contact-transmitted diseases.

## Figures and Tables

**Figure 1 foods-12-02701-f001:**
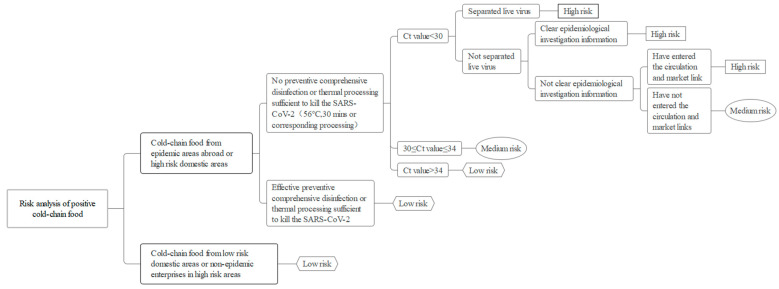
The risk assessment of positive cold-chain food.

**Figure 2 foods-12-02701-f002:**
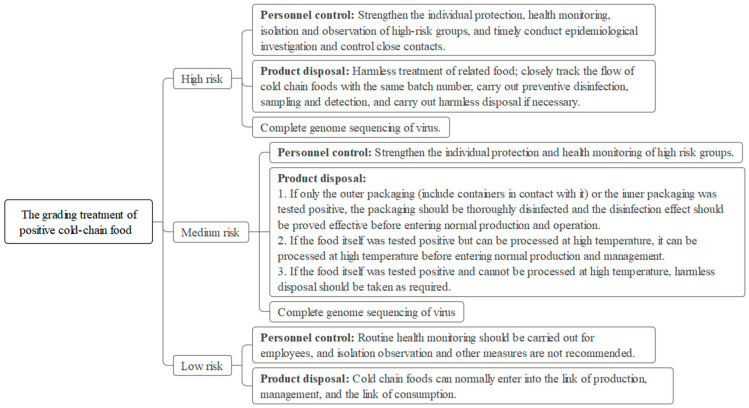
The grading treatment of positive cold-chain food.

**Figure 3 foods-12-02701-f003:**
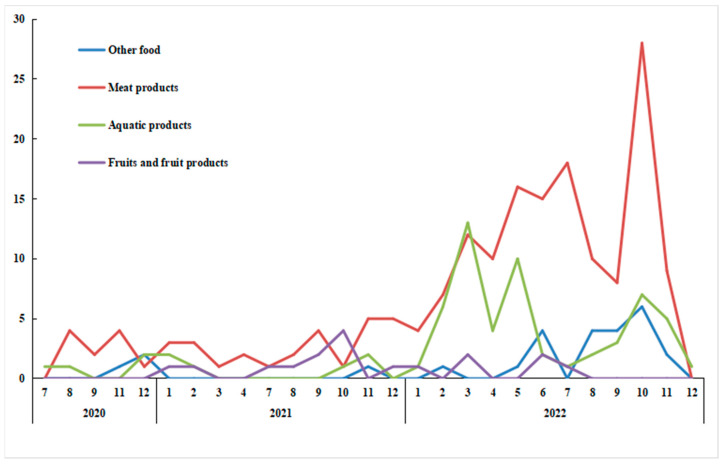
Temporal trends in positive cold-chain events.

**Table 1 foods-12-02701-t001:** Regional sources of positive events and samples.

The Source Area of Positive Sample	No. of Events (%)	No. of Positive Samples (%)
Asia	91 (32.16)	329 (36.39)
South America	71 (25.09)	209 (23.12)
North America	60 (21.2)	167 (18.47)
Europe	52 (18.37)	165 (18.25)
Australia	4 (1.41)	19 (2.1)
Africa	5 (1.77)	15 (1.66)
Total	283	904

**Table 2 foods-12-02701-t002:** Location along the food chain of positive events and samples.

Type of Discovery	Other Food		Meat Products		Aquatic Products		Fruits and Fruit Products		Total	
	No. Events (%)	No. of Positive Samples (%)	No. of Events (%)	No. of Positive Samples (%)	No. of Events (%)	No. of Positive Samples (%)	No. of Events (%)	No. of Positive Samples (%)	No. of Events (%)	No. of Positive Samples (%)
Centralized supervision cold storage	23 (88.46)	80 (96.39)	158 (90.29)	494 (96.67)	58 (89.23)	269 (97.46)	9 (52.94)	24 (70.59)	248 (87.63)	867 (95.91)
Market	1 (3.85)	1 (1.20)	8 (4.57)	8 (1.57)	2 (3.08)	2 (0.72)	6 (35.29)	8 (23.53)	17 (6.01)	19 (2.10)
Downstream cold storage	2 (7.69)	2 (2.41)	7 (4.00)	7 (1.37)	4 (6.15)	4 (1.45)	1 (5.88)	1 (2.94)	14 (4.95)	14 (1.55)
Interception of freight	0 (0.00)	0 (0.00)	1 (0.57)	1 (0.20)	1 (1.54)	1 (0.36)	1 (5.88)	1 (2.94)	3 (1.06)	3 (0.33)
Smuggle	0 (0.00)	0 (0.00)	1 (0.57)	1 (0.20)	0 (0.00)	0 (0.00)	0 (0.00)	0 (0.00)	1 (0.35)	1 (0.11)
Total	26 (9.19)	83 (9.18)	175 (61.84)	511 (56.53)	65 (22.97)	276 (30.53)	17 (6.01)	34 (3.76)	283	904

**Table 3 foods-12-02701-t003:** Positive events and samples associated with various sampling surfaces.

Sampling Position	Product Classification	No. of Events (%)	No. of Positive Samples (%)
Outer packing	Other food	26 (9.56)	83 (9.37)
	Meat products	169 (62.13)	501 (56.55)
	Aquatic products	64 (23.53)	274 (30.93)
	Fruits and fruit products	13 (4.78)	28 (3.16)
	Total	272	886
Inner packing	Meat products	7 (63.64)	17 (62.96)
	Aquatic products	2 (18.18)	8 (29.63)
	Fruits and fruit products	2 (18.18)	2 (7.41)
	Total	11	27
Product surface	Meat products	3 (42.86)	7 (50.00)
	Aquatic products	1 (14.29)	2 (14.29)
	Fruits and fruit products	3 (42.86)	5 (35.71)
	Total	7	14
Environmental surface	Meat products	1	7

**Table 4 foods-12-02701-t004:** Time distribution of events by risk level.

Year	Month	Low Risk	Medium Risk	High Risk	Total
2020	July	1 (100)	0 (0)	0 (0)	1
	August	5 (100)	0 (0)	0 (0)	5
	September	2 (100)	0 (0)	0 (0)	2
	October	5 (100)	0 (0)	0 (0)	5
	November	5 (100)	0 (0)	0 (0)	5
	December	0	0	0	0
2021	January	4 (66.67)	2 (33.33)	0 (0)	6
	February	5 (100)	0 (0)	0 (0)	5
	March	1 (100)	0 (0)	0 (0)	1
	April	2 (100)	0 (0)	0 (0)	2
	May	0	0	0	0
	In June	0	0	0	0
	July	2 (100)	0 (0)	0 (0)	2
	August	2 (66.67)	1 (33.33)	0 (0)	3
	September	5 (83.33)	1 (16.67)	0 (0)	6
	October	4 (66.67)	2 (33.33)	0 (0)	6
	November	2 (25)	5 (62.5)	1 (12.5)	8
	December	5 (83.33)	1 (16.67)	0 (0)	6
2022	January	5 (83.33)	1 (16.67)	0 (0)	6
	February	13 (92.86)	1 (7.14)	0 (0)	14
	March	15 (55.56)	12 (44.44)	0 (0)	27
	April	12 (85.71)	2 (14.29)	0 (0)	14
	May	25 (92.59)	2 (7.41)	0 (0)	27
	In June	19 (82.61)	4 (17.39)	0 (0)	23
	July	18 (90)	2 (10)	0 (0)	20
	August	11 (68.75)	5 (31.25)	0 (0)	16
	September	11 (73.33)	4 (26.67)	0 (0)	15
	October	35 (85.37)	6 (14.63)	0 (0)	41
	November	11 (68.75)	5 (31.25)	0 (0)	16
	December	1 (100)	0 (0)	0 (0)	1
Total		226 (79.86)	56 (19.79)	1 (0.35)	283

**Table 5 foods-12-02701-t005:** Examination of personnel, environment, and imported products in centralized supervision cold storage 2021–2022.

		Baiyun District		Panyu District		Huangpu District		Nansha District	Total
		2021	2022	2021	2022	2020	2021	2022	
Personnel sampling	Workers in high-risk positions	4092	4185	21,537	64,122	2076	21,198	13,694	130,904
	Workers in low-risk positions	7440	7440	10,510	19,140	2420	26,834	2178	75,962
Product sampling	Meat products	950	3305	336,264	255,470	13,540	38,291	51,483	699,303
	Aquatic products	1403	3180	28,857	30,452	13,462	60,117	7860	145,331
	Fruits and fruit products	930	4411	1527	9528	0	0	3402	19,798
	Other food	514	3643	18104	41575	4534	35,460	4476	108,306
Environmental sampling		567	600	1618	8634	278	1410	1789	14,896
Imported products	Number of shipments of imported goods	2113		14,855	17,895	11514	8669	3822	58,868
	Food imports (tons)	10,518.4	10,518.4	371,865.3	447,106.7	38,847.1	142,330.7	91,292.4	1,112,479.0

Note: The centralized supervision cold-storage facilities of Guangzhou are distributed in Baiyun, Panyu, Huangpu, and Nansha districts. Two facilities are in Huangpu District, and one each is located in Panyu, Nansha, and Baiyun districts. The facility in Nansha District was operationalized as recently as 2022. The individual monitoring data are the total number of personnel monitoring; the monitoring frequency is three times weekly and twice weekly for high-risk and low-risk personnel, respectively. The frequency of environmental monitoring is once weekly. PCR testing was conducted for every shipment of imported goods.

## Data Availability

Data is contained within the article.

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
