# Peer review of "Cold-Chain-Food-Related COVID-19 Surveillance in Guangzhou between July 2020 and December 2022"

_foods, 2023, doi:10.3390/foods12142701_

Round 1

Reviewer 1 Report

The work analyzed surveillance data on Cold-chain food-related COVID-19 in Guangzhou between July 2020 and December 2022.

The topic is interesting, and the introduction is straightforward and presents relevant background information.

I recommend the reconsideration of the paper after satisfactorily addressing the appended comments.

(1)   The manuscript can benefit from careful proofreading and improved use of the English language.

(2)   The use of the word “surveillance” in the abstract is incorrect. Authors should use the verb “surveil” or a more suitable synonym such as “examine”.

(3)   Line 71: “Among table surface” has an uncertain meaning

(4)   Figure 1 seems to cover the risk assessment aspect, so it should be revised as “Risk assessment of positive cold-chain foods” Risk analysis contained other vital components not addressed in this work.

(5)   What was the status of the cold food chain food safety management in the investigated area before COVID-19? This background information is very vital to include in the introduction.

(6)   What are the food safety management systems being enforced in those areas? Was there strict compliance with those standards before covid-19?  

(7)   I know the test kit in China for covid test appears to give reliable results, but there may be chances of a good proportion of “false positives or false negatives”. So how do your methods ensure the accuracy of the data?

(8)   The authors should mention the limitations (if any) of the current investigations. 

NA

Author Response

Dear reviewer:

Please see the attachment,Thank you!

Reviewer 2 Report

What is the objective of this paper? It was not mentioned in the Introduction, but just briefly in the Abstract. The paper also lacks a Conclusion, although some traits of it can be elucidated in the discussion.

How many samples were tested in total? Figures are a bit incomprehensible and the claim that the "positive detection rate is low" as stated in line 259 is unsupported due to poorly structured data. This should be more concise and straightforward.

Line 38: Cold chain is for sure not the cause, it merely might be considered a route.

Line 59: Reference missing and more detail needed.

Line 78: The application of disinfectants is surely not considered sterilization.

Line 181: Do not repeat data already mentioned just before the table.

Line 195: I think there is a mistake in presenting figures. 272 cases were pertinent to the positive outside packaging and that clearly is missing here.

Line 284: Goods are not infected (non-viable substances) but contaminated.

Some editing for English language is required throughout the manuscript due to too many mistakes

Author Response

(The authors gave the same response as above.)

Round 2

Reviewer 2 Report

Thank you for making corrections and revision of problematic sections. Changes you made are satisfactory and I suggest paper to be accepted.